# Could FeNO Predict Asthma in Patients with House Dust Mites Allergic Rhinitis?

**DOI:** 10.3390/medicina56050235

**Published:** 2020-05-14

**Authors:** Ioana Adriana Muntean, Ioana Corina Bocsan, Stefan Vesa, Nicolae Miron, Irena Nedelea, Anca Dana Buzoianu, Diana Deleanu

**Affiliations:** 1Department of Allergology and Immunology, “Iuliu Hatieganu” University of Medicine and Pharmacy, 400162 Cluj Napoca, Romania; Adriana.Muntean@umfcluj.ro (I.A.M.); irena.nedelea@umfcluj.ro (I.N.); diana.deleanu@umfcluj.ro (D.D.); 2Toxicology and Clinical Pharmacology, Department of Pharmacology, “Iuliu Hatieganu” University of Medicine and Pharmacy, 400337 Cluj Napoca, Romania; stefan.vesa@umfcluj.ro (S.V.); abuzoianu@umfcluj.ro (A.D.B.); 3Department of Clinical Immunology, Sahlgrenska University Hospital, 41346 Göteborg, Sweden; nicolae.miron@vgregion.se

**Keywords:** allergic rhinitis, allergic inflammation, asthma, FeNO, house dust mites

## Abstract

*Background and Objectives*: The evolution of allergic rhinitis to asthma is a part of “atopic march”. The aim of this study was to analyze possible predictive markers for asthma occurrence in patients with allergic rhinitis to house dust mites (HDM). *Materials and Methods*: Fifty-eight patients with persistent allergic rhinitis (PAR) were included. The clinical, biological evaluation and fractionated exhaled nitric oxide (FeNO) measurement were performed at enrolment. The patients were clinically evaluated after one year to determine asthma occurrence. *Results:* The severity of rhinitis symptoms, levels of total immunoglobulin E (IgE), ICAM-1, VCAM-1, E-selectin and IL-6, but not IL-8 and TNF-α were higher in patients with allergic rhinitis who developed asthma compared to non-asthmatics, but the differences were not significant to considered them as predictive factors for asthma occurrence. The risk of asthma was independently influenced by patients aged over 30 years ((OR-3.74; CI95% 0.86–16.31; *p* = 0.07), a duration of allergic rhinitis over 12 months ((OR-4.20; CI95% 0.88–20; *p* = 0.07) and a basal FeNO over 28 parts per billion (pbb) ((OR-18.68; CI95% 3.79–92.05; *p* < 0.001). *Conclusion*: Clinical and biological parameters may predict asthma occurrence in patients with persistent allergic rhinitis to HDM. Adult patients with a longer duration of rhinitis symptoms and a high level of FeNO have a greater risk to develop asthma.

## 1. Introduction

Allergic rhinitis is the most frequent IgE-mediated disease and its prevalence is increasing in the last decades [1,2]. Allergic rhinitis is a risk factor for asthma development and may be clinically relevant before or after asthma diagnosis [3]. Allergic inflammation is the key of understanding these diseases and the evolution of allergic rhinitis to asthma [2,4,5].

Allergen exposure leads to mast cell degranulation in nasal mucosa and the release of mediators, mainly histamine and leukotrienes. Cytokines released from Th2 lymphocytes are responsible for inflammatory cells recruitment in affected tissues via adhesion molecules, like E-selectin, ICAM and VCAM. The recruited cells, eosinophils, neutrophils and Th2 lymphocytes, are responsible for producing more proinflammatory cytokines (IL-1β, TNF-α, IL-3, IL-4, IL-5, IL-6 and IL-8), augmenting the inflammation in airways and injury through formation of toxic reactive nitrogen species [5,6,7,8].

The evolution from allergic rhinitis to asthma was reported in several studies, and it is a part of “atopic march” [2,9,10,11]. Allergic rhinitis is a risk factor for developing asthma [12], especially if the onset is severe and occurs in childhood [13]. Those two diseases often coexist and represent “a single airways allergic disease” [2,9]. This concept of “united airway disease” raises the question “Which patients with allergic rhinitis will develop asthma?”

The relationship between upper and lower airway inflammation is not completely understood yet. Genetic studies show that asthma and allergic rhinitis partly coexist because they share many genetic risk variants that dysregulate the expression of immune-related genes [14]. But not only genetic factors are important, environmental ones (allergens exposure) might also contribute to this evolution. The concept of a “united airway disease” could be explained through the migration of inflammatory cells and mediators from nasal secretions to lower airways, by inhalation and aspiration, acting as triggers of inflammation in the lower part [9,15,16]. Other additional mechanisms along with inflammation might contribute to asthma occurrence, including nasal bronchial reflex and alteration of physical filter function of the nose, which can induce bronchial hyperreactivity even in non-atopic patients [16,17,18].

The clinical aspect of asthma is variable, from a classical description (chest tightness, dyspnea, wheezing and cough) to only chronic cough or dyspnea to mild physical effort [7,15]. Some patients with allergic rhinitis may present rare, mild asthma symptoms, which are not related to rhinitis severity and could be actually a matter of perception of asthma symptoms, like dyspnea [7].

FeNO is a marker of lower eosinophilic inflammation in allergic diseases, especially in asthma. FeNO measurement is used for asthma diagnosis, to differentiate its phenotype and to monitor treatment response [19,20]. In asthmatic patients, a high nitric oxide is more correlated with the risk of having an asthmatic access rather than a predisposition to have asthma [21]. In patients with allergic rhinitis, measurement of FeNO might also indicate the presence of eosinophilic inflammation and might predict the development of lower airway symptoms [19,22].

The aim of this study was to investigate the risk of asthma development in patients with persistent allergic rhinitis to house dust mites after one year and the role of inflammatory cytokines, adhesion molecules and FeNO in predicting asthma occurrence.

## 2. Materials and Methods

### 2.1. Study Design, Site and Ethical Approval

The present study is a post-hoc analysis of an initial randomized control trial (RCT) that included patients with persistent allergic rhinitis [23]. The present research analyzed clinical and biological factors that might predict asthma in patients with allergic rhinitis to HDM. A diagnosis of allergic rhinitis was established according to international guidelines, based on history, clinical evaluation and the skin prick test (SPT) [24].

Fifty-eight patients with persistent allergic rhinitis to HDM (median age 27.5 (23–37) years and sex ratio M:F = 1:1), that were evaluated in Allergology Department, were included in the present analysis. The study protocol was approved by University of Medicine and Pharmacy Ethics Committee (approval no. 535/02.09.2009) and all patients signed the informed consent before enrollment. The study protocol and clinical evaluation was similar to initial RCT [23]. The exclusion criteria were as follows: the presence of asthma or nasal polyps, acute and chronic upper respiratory infections, administration of intranasal or systemic corticosteroids or H1 antihistamines in the previous 30 days. The initial evaluation was performed between February 2009 and November 2011.

### 2.2. Clinical Evaluation

From anamnesis, we noted the following demographic data: age, sex, living area (urban/rural) and the duration of allergic rhinitis symptoms prior enrollment. Retrospectively, for 12 h, we evaluated the allergic rhinitis’ symptoms (rhinorrhea, nasal congestion, sneezing, nasal and ocular itching), and their severity on a scale from 0 to 3 (0 = absent, 1 = mild, 2 = moderate, 3 = severe). At the end we calculated the total symptoms score (TSS). Based on TSS values we differentiated between mild allergic rhinitis (TSS < 6) and moderate–severe allergic rhinitis (TSS 6).

As we previously mentioned, patients that presented low airways symptoms (dyspnea, cough and wheezing) associated to specific symptoms of allergic rhinitis were excluded from the present analysis. The patients also completed an ENT examination to exclude a possible nasal obstruction of other cause or nasal polyps. Patients with nasal polyps or another ENT disease were also excluded.

After one year, we repeated the clinical evaluation to determine the possible development of asthma. We noticed the occurrence of asthma symptoms (cough, wheezing, dyspnea) or an asthma exacerbation that required specific treatment in this period of time.

Spirometry was performed at enrollment in order to exclude a possible impaired lung function due to asthma presence and after one year. We considered asthma development if one of these clinical or functional criteria were present in the period of one year.

### 2.3. Skin Prick Tests (SPT)

The atopy diagnosis was established through a skin prick test at enrollment, according to international guideline [25]. The skin prick test included the following panel of allergens: house dust mites (Derm. Pteronyssinus (Der p) and Derm. Farinae (Der f)), pollens (grasses, cereals, birch and weeds), animal dander (cat and dog) and molds (Alternaria alternata). Standardized allergen extracts (Hal Allergy, Netherlands) were used. The value in mm was recorded as a medium diameter wheal size.

### 2.4. FeNO Measurement

Fractionated exhaled nitric oxide (FeNO) was measured at enrollment, according to international recommendations [26], using NIOXMINO^®^ (Aerocrine, Sweden). The measured values were expressed in parts per billion (ppb). A standardized value of 25 ppb was considered as normal upper limit.

### 2.5. Biological Evaluation

All the biological parameters were determined at the beginning of the study. Total IgE plasma level was done using the electrochemiluminescence immunoassay method (ECLIA). The obtained values were expressed as UI/mL, considering a normal value <100 UI/mL. The eosinophils (Eo) were manually counted from peripheral blood on a slide and their value was expressed as %. We considered a normal value between 2%–4%.

Plasma levels of adhesion molecules (ICAM-1, VCAM-1 and E-selectin) and cytokines (TNF-α, IL-6 and IL-8) were determined at the initial visit. Five milliliters of blood sample was collected and centrifuged within the first hour, followed by serum separation. The obtained serum was stored at −80 °C until the determination was performed. The plasmatic levels of all inflammatory markers were determined by sandwich ELISA technique using an ELISA reader StatFax 303. All the aforementioned determinations were done according to the manufacturers’ instructions, using ELISA kits from Quantikinine R&D system, USA. For each assay, samples were prepared according to instructions and protein levels were calculated based on four-parameter logistic (4-PL) curve fit.

### 2.6. Statistical Analysis

Statistical analysis was carried out using the MedCalc Statistical Software version 18.10 (MedCalc Software bvba, Ostend, Belgium; http://www.medcalc.org; 2018). Quantitative data were evaluated for normality of distribution and variables with abnormal distribution were characterized by median and 25–75 percentiles. Qualitative data were expressed as frequency and percent. Comparisons between groups were performed using the Mann–Whitney (for quantitative data) and chi-square tests (for qualitative data). Spearman rho coefficient was used for examining the correlation between variables. ROC curves were used in order to find out cut-off values for quantitative variables that could discriminate between patients with asthma and those without. A multivariate binary logistic regression was used for assessment of independent contribution of variables that achieved statistical significance in univariate analysis for asthma onset. A *p* value < 0.05 was considered statistically significant.

## 3. Results

From the entire group of patients with allergic rhinitis to HDM, 21 patients (36.2%) developed asthma after one year of surveillance.

Patients’ demographic, clinical and biological data are presented in Table 1.

Analyzing demographic data, we noticed that asthma occurrence is correlated with patients’ age, but not with their gender or living area (see Table 1). More male patients developed asthma compared to females, but the difference was not statistically significant.

Forty patients (68.9%) presented persistent moderate–severe forms of allergic rhinitis, proved by an initial TSS over 6 (median 8.5 (5–11)). The development of asthma was not correlated with a moderate–severe form of allergic rhinitis (*p* = 0.5), even if more patients with asthma had previously moderate–severe allergic rhinitis (76.2% vs. 64.9%). Initial TSS of allergic rhinitis was higher in patients with asthma, but the difference was not statistically significant. Initial was not correlated with the duration of AR and was not influenced by patients’ sex or living area (*p* > 0.05). The duration of allergic rhinitis is significantly higher in patients with asthma after one year of surveillance.

The markers of allergic inflammation, total IgE and eosinophils were higher in patients with asthma after one year, compared with patients with allergic rhinitis without asthma, but the differences were not statistically significant (*p* > 0.05). We noticed similar results for adhesion molecules (ICAM-1, VCAM-1 and E-selectin) and IL-6, but not for TNF-α and IL-8. Only FeNO was significantly higher in patients with allergic rhinitis and asthma compared to those without asthma (see Table 1). The initial values of biological markers were not influenced by patients’ age, sex and living area, duration or severity of allergic rhinitis (*p* > 0.05).

Thirty-seven patients (63.8%) were polysensitized to both indoor and outdoor allergens, but the symptoms of rhinitis were present after exposure to HDM, while 21 patients (36.2%) were sensitized only to HDM. All the patients were sensitized to Der p, while 86.20% (50 patients) were sensitized also to Der f. The wheal size of Der p sensitization was higher in patients who developed asthma compared with those without asthma. The asthma development was not correlated with number or type of sensitizations to other allergens, except HDM (*p* > 0.05).

In patients with allergic rhinitis to HDM, we observed a moderate positive correlation between baseline values of TNF-α and IL-8 (r = 0.327, *p* = 0.01), IL-6 and IL-8 (r = 0.437, *p* = 0.001), weak negative correlation between VCAM-1 and IL-8 (r = −0.290, *p* = 0.03) and a weak positive one between TNF-α and IL-6 (r = 0.260, *p* = 0.04). 

The ROC curve for patients’ age, duration of allergic rhinitis and FeNO were analyzed and the cut-off values were calculated for these parameters in relation with asthma onset after one year of the inclusion visit. The cut off values, AUC, sensitivity and specificity are presented in Table 2.

In order to find out which parameter was independently associated with asthma’ occurrence in patients with allergic rhinitis to HDM, we used a multivariate logistic regression (Table 3). Variables which achieved statistical significance in univariate analysis were introduced in the regression. Our model explained 36.2% of asthma prevalence. Age and allergic rhinitis duration were very close to statistical significance, probably due to the small number of patients. FeNO > 28 ppb was the only independent variable that predicted the onset of asthma at a one-year follow-up (Table 3).

## 4. Discussion

The present study showed a significant association of asthma symptoms in patients with persistent allergic rhinitis to HDM after one year of surveillance, 36.2% of them presenting asthma. Clinical (duration, patient age) and biological data (inflammatory markers) may predict asthma development. FeNO was an independent variable that predicted the onset of asthma at one-year follow-up.

Allergic rhinitis and asthma are considered a single respiratory disease involving two parts of the airways [2,9]. In the present study, the authors found a prevalence of asthma of 36.2% after a one-year follow-up. Similar data were already reported in the literature, with a variable prevalence between 20% and 50% of patients with allergic rhinitis [9,10]. The prevalence found in the present research was higher compared to the study published in 1998, where the prevalence of asthma in patients with allergic rhinitis was lower (21.3%) after 23 years of follow up [27]. But Greisner et al. [27] included all patients with allergic rhinitis at the same age (first year of faculty), not only patients with persistent forms and different ages. The patients from the present research had a median age of 27.5 year, higher than the age of patients from the Greisner study. A similar prevalence of asthma was also established in children (30% in 13–14 years group and 35% in 6–7 years group) [28], but they did not follow prospectively their patient and the prevalence was established retrospectively.

The current diagnose of rhinitis relies on combination of three types of data: historical, clinical examination, and allergy diagnostic testing, which allows differentiation into three subgroups: allergic, infectious, and non-allergic non-infectious rhinitis [24,29]. Bronchial hyperreactivity is commonly present in patients with persistent moderate severe allergic rhinitis and should be suspected if other risk factors are present (allergen and viral exposures, indoor and outdoor pollution, allergic rhinitis duration and severity) [30,31,32]. The validation of some clinical and biological factors will permit to phenotype and endotype allergic rhinitis in order to find a form of “asthma risk” allergic rhinitis. Maybe a different approach of allergic rhinitis according to its phenotype and endotype could be done in order to prevent asthma development.

In this study, the clinical, biological and inflammatory markers that might influence the appearance of asthma in patients with allergic rhinitis to HDM were evaluated. The authors included only patients with persistent allergic rhinitis in this study, knowing that duration of symptoms and their severity are risk factors for asthma development [30,32]. A duration of allergic rhinitis over 12 months was considered a risk factor for asthma occurrence similar to other previous studies, in both adults and children [3,27]. The severity of disease was not correlated with asthma occurrence in this research. In Bousquet et al.’s and del Curvillo et al.’s studies [30,32], the severity of rhinitis was correlated with asthma development, but they included patients with both intermittent and persistent allergic rhinitis, while in the present research all the patients had persistent forms.

Rhinitis phenotypes were also described in relationship to sensitization pattern [33]. Sensitization to HDM is a risk factor for asthma development because they are perennial allergens [9]. In this study, only patients with sensitization to HDM were included. In addition, the sensitization may influence in different degree the severity of allergic rhinitis and the evolution to asthma, as Vidal’s study already mentioned [33]. Vidal et al. reported that severe allergic respiratory disease was associated with higher levels of both total IgE and specific IgE to HDM. The presence of sIgE to both Der p 1 and Der p 2 was associated with asthma among HDM-allergic patients [33]. Our results revealed an almost significant association (*p* = 0.052) in univariate analysis between the size of the skin prick test to Der p and the occurrence of asthma. The mean size of wheal to asthmatic patient was 7.89 mm, which may confirm the presence of clinically manifested symptoms as for both asthma and rhinitis as in Haahtela’s study [34]. The authors of this research did not determine the level of sIgE to HDM; they confirmed the sensitization based only on the skin prick test. Probably a large number of patients may give more information regarding the role of wheal size in assessing rhinitis or asthma symptoms.

An important step in implementing the precision medicine in patients with allergic rhinitis is to identify possible biomarkers which characterize the endotypes and may guide us to different therapeutic approaches. In asthma research, the endotypes are already described, based on different biomarkers (cytokines, cells). Allergic rhinitis might have complex endotypes and the current understanding of cellular and molecular processes may lead to identify certain biomarkers that characterize the endotypes, but studies are still required to confirm them. There are several modulators of endotype expression, such as environment, microbiome, lifestyle and nasal anatomy. In several studies, the following endotypes of rhinitis were proposed: type 2 immune response rhinitis, type 1 immune response rhinitis, neurogenic rhinitis, epithelial dysfunction [26,35,36].

However, in the present study, the authors focused only on type 2 immune response allergic rhinitis with HDM sensitization alone or with other co-sensitizations. Previous studies revealed that some markers are increased in patients with allergic rhinitis to HDM: total serum IgE, blood and intranasal eosinophils, various types of cytokines [18,37] and adhesion molecules [6,38,39]. In this study, the comparative univariate analysis of patients with allergic rhinitis with or without asthma after a one-year surveillance showed that mean values of adhesion molecules and IL-6 were increased in the first subgroup compared to patients without asthma, but the differences did not reach the statistical significance and, because of this, they were not included in multivariate analysis. The plasmatic values of cytokines are extremely variable and more patients should be included in order to confirm them as biomarkers. Additionally, their plasmatic level should be correlated with their level in nasal or bronchial lavage. A previous study in children reported that a cytokine imbalance predicted asthma occurrence after three years of surveillance [40], but they concluded that respiratory viral infections may play a significant role in this imbalance, not only atopy. Therefore, these inflammatory markers should be evaluated in different population subgroups, adults and children, and in conjunction with other risk factors, in order to confirm their potential role as biomarkers.

As a biologic marker, only FeNO was significantly higher in patients with allergic rhinitis and asthma compared to those without asthma. Patients with allergic rhinitis to HDM that developed asthma have an increased systemic inflammation, which was correlated with subclinical inflammation in the lower airways. FeNO is a biomarker of atopy and eosinophilic airway inflammation [41]. Previous studies found a correlation of FeNO level and type of sensitization, in both adults and children. Jouaville et al. found that FeNO level increased with the number of positive skin prick tests (SPTs) in both asthmatics and non-asthmatic subjects, but in cases of an equal level of positive SPTs the FeNO was higher in asthmatic than in non-asthmatic, reflecting its role in asthma diagnoses [42]. The exact role of FeNO in the prediction of asthma in patients with allergic rhinitis is still unclear.

In the present study, patients with allergic rhinitis to HDM with continuous symptoms for more than 12 months and an elevated FeNO at presentation over 28 ppb had 18-fold higher risk for asthma after one year. Malinovschi et al. used FeNO in evaluation of allergic rhinitis persistence and severity in general population of adolescents, but the author did not calculate a cut off value which may predict the persistence of rhinitis [43]. Additionally, they evaluate the persistence of allergic rhinitis symptoms, not the occurrence of asthma ones. Di Cara et al. found that children with allergic rhinitis and elevated values of FeNO over 35 ppb developed asthma after five years of monitoring [44]. Similar results were also reported in children and adults [45,46]. The cut off value in our study was lower than in the Di Cara study [44], but only adults were included, while the Di Cara study included only children.

The factors that lead to asthma in patients with allergic rhinitis are multiple. Further studies are still needed to evaluate the evolution of allergic inflammation. However, from the clinical practice point of view, it is important to evaluate all patients with allergic rhinitis for asthma symptoms [47]. Patients with allergic rhinitis and other risk factors for asthma should be carefully evaluated for the presence of subclinical inflammation in the lower airways, which is the substrate for bronchial hyperreactivity.

This study has a strong value because it emphasized the role of FeNO, a marker of lower eosinophilic inflammation in allergic rhinitis in order to predict the evolution of disease to other manifestations. There are some limitations of this study. Firstly, a small number of patients were included in the study. Secondly, the surveillance period is short and some of the investigated markers may not be able to predict the asthma development so soon. It could be interesting to also evaluate the modulation of these biological markers under different treatments that are recommended in allergic rhinitis. The authors evaluated the markers in patients with rhinitis induced by HDM, but they could not compare patients with monosensitization with the ones polysensitized because of the small number of subjects included in the present analysis.

## 5. Conclusions

Clinical and biological parameters may predict asthma occurrence in patients with persistent allergic rhinitis to house dust mites. Adult patients with a longer duration of rhinitis symptoms and a high level of FeNO over 28 have a greater risk to develop asthma. The severity of symptoms and the serum inflammatory cytokines and adhesion molecules are not correlated with asthma occurrence after one year of monitoring. FeNO could become a useful biomarker in predicting a specific endotype of allergic rhinitis with high risk to develop asthma.

## Figures and Tables

**Table 1 medicina-56-00235-t001:** Comparison between patients based on asthma diagnosis at one-year follow-up.

Variable		Total (*n* = 58)	Patients with Asthma (*n* = 21)	Patients without Asthma (*n* = 37)	*p*
Age (Years) *		27.5 (23–37)	33 (24.5–40)	26 (22–31)	0.014
Sex ^	M	50% (29)	57.1% (12)	45.9% (17)	0.5
	F	50% (29)	42.9% (9)	54.1% (20)	
Living area ^	U	82.8% (48)	85.7% (18)	81.1% (30)	0.7
	R	17.2% (10)	14.3% (3)	18.9% (7)	
Onset of AR symptoms (months) *		24 (6–60)	36 (15–66)	12 (3–48)	0.04
Total symptom score *		8.5 (5–11)	9 (5.5–13)	8 (5–11)	0.2
FeNO (ppb) *		24 (16–46)	45 (30.5–68)	19 (16–28)	<0.001
Total IgE (UI/l) *		106.5 (44.55–201.5)	118 (35.4–293)	104 (49.8–233)	0.9
Eosinophils *		0.05 (0.026–0.071)	0.05 (0.02–0.08)	0.04 (0.02–0.06)	0.5
E selectin (ng/mL) *		3.28 (2.35–4.69)	3.45 (2.26–4.81)	2.32 (2.43–4.64)	0.9
ICAM (ng/mL) *		21.53 (18.88–26.55)	22.98 (20.02–27.18)	20.87 (18.39–24.32)	0.1
VCAM (ng/mL) *		48.53 (40.95–59.50)	56.38 (41.34–60.55)	46.81 (40.93–55.95)	0.1
TNF-α (pg/mL) *		1.78 (1.22–2.33)	1.62 (1.01–2.11)	1.93 (1.34–2.41)	0.2
IL-6 (pg/mL) *		1.05 (0.75–1.70)	1.25 (0.71–1.84)	1.05 (0.78–1.65)	0.6
IL-8 (pg/mL) *		5.32 (3.33–9.38)	5.05 (2.35–8.20)	5.32 (3.95–9.51)	0.2
Wheal size of Der p allergen at prick test (mm)		7.23 ± 2.92	7.89 ± 3.24	6.57 ± 2.61	0.052

Data are expressed as * median; 25–75th percentile; ^ Data are expressed as %, *n*; Tests used: Mann–Whitney (for quantitative data) and chi-square tests (for qualitative data); Significance *p* < 0.05. Abbreviations: AR, allergic rhinitis; F, female; FeNO, fractional exaled nitric oxide; M, male; R, rural; TSS, total symptoms score; U, urban.

**Table 2 medicina-56-00235-t002:** ROC curve analysis for asthma diagnosis at 1-year follow-up.

Parameter	AUC	Cut-Off Value	Sensitivity	Specificity	*p*
Age	0.696 (95%CI 0.56–0.81)	>31 years	61.90% (95%CI 38.4–81.9%)	78.38% (95%CI 61.8–90.2%)	0.007
Duration of AR	0.659 (95%CI 0.52–0.77)	>12 months	76.19% (95%CI 52.8–91.8%)	54.05% (95%CI 36.9–70.5%)	0.02
FeNO	0.79 (95%CI 0.66–0.88)	>28 ppb	85.71% (95%CI 63.7–97.0%)	78.38% (95%CI 61.8–90.2%)	<0.001

Abbreviations: AR, allergic rhinitis; AUC, aria under de curve; CI, interval of confidence; FeNO, fractional exhaled nitric oxide; parts per billion.

**Table 3 medicina-56-00235-t003:** Multivariate analysis for asthma occurrence in patients with allergic rhinitis to HDM.

Variables	B	*p*	OR	95% C.I. for OR	
				Min	Max
Age > 31 yo	1.321	0.07	3.746	0.860	16.311
Duration of AR > 12 months	1.437	0.07	4.209	0.885	20.008
FeNO > 28 ppb	2.928	<0.001	18.682	3.791	92.057
Constant	−0.825	0.04	0.438		

Abbreviations: AR, allergic rhinitis; B, CI, interval of confidence; FeNO, fractional exhaled nitric oxide; HDM, house dust mites; Max, maximum; Min, minimum; male; OR, odds ratio; ppb, parts per billion. Test used: binary logistic regression.

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
