# Peer review of "Could FeNO Predict Asthma in Patients with House Dust Mites Allergic Rhinitis?"

_medicina, 2020, doi:10.3390/medicina56050235_

Round 1

Reviewer 1 Report

This is an interesting study, albeit relatively small-scale, retrospectively analysing data from a RCT  study. The publication adds to the current data on the relationship and natural history between persistent allergic rhinitis  and asthma.

I have made a few comments where the English/text may be improved to aid in comprehension and some comments about the information presented.

  1. Abbreviations of PAR and FeNO need to introduced early in the text after their description in full. e.g. FeNO is used several times in the introduction but only described in fill on line 104. PAR and FeNO are used in the abstract without full description, unlike HDM.
  2. line 61. diagnosis not diagnose
  3. line 78.  Incusions/exclusions "were similar". This appears a little undefined.  Were they identical with additional exclusion as noted subsequently? I note that lines 87-88 and 89-90 and 94 suggest exclusons of  low airways symptoms, nasal obstrutions and impaired lung function by spirometry on enorllment? Is this the case?
  4. line 98. atopy diagnosis not diagnose
  5. Lines 109 and 114. plasma not plasmatic
  6. "Quantitative data were evaluated for normality of distribution and were characterised by median and 25-75 percentiles".  The authors state they used non-parametric tests for correlations between variables. Were non-parametric comarisons used for the data in table 1 and the multivariate analyses (table 3)?  The authors could be more explicit, given they did introduce " normality of distribution". 
  7. Line 129 lose the a before cut-off values, 
  8. Line 134 developed not develop
  9. It may help readers evaulate those parameters close to statistical significance if analytical precision of the methods (e.g. ICAM, VCAM, E-selection, TNF, IL6 & 8) can be provided, as analytical precision will add the the biological variation in such parameters and weaken their possible prognostic value.
  10. Table 1 wheal mm and lines 223-227.  Are the wheal sizes for asthma/non-asthma categories in table 1 the right way around? the text and the table appear at odds.  What was the prick test antigen described in  table 1? Line 225 suggests Der p 1 and Der p 2 but the methods section for SPT suggests Derm Pterony. a Derm. Farinae?  The method section states skin prick test, but I presume a panel of skin prick tests were employed using the allergens individually as described? Maybe the authors could review and tighten up text about SPT in methods, results, table  and discussion.
  11. Table 1. Onset of AR symptoms (months). Is this duration of AR symptoms prior to enrollment? Duration is used in the text elsewhere. 
  12. Lines 158-161. These are potentially important findings. Could the numbers and statistical analysis on monosentisised (i.e. HDM) versus poly-sensitisation be incorporated into table 1 to give greater clarity?
  13. Table 2 headings Sensitivity not Sensibilty.
  14. Line 192. After Greisner et al, add [24].
  15. Paragraphs 199-207 and 231-254. We have the concepts of phenotype and endotype introduced. I am more conversant with the former.  Is the latter, correct and necessary usage?
  16. Line 266. risk not risks 
  17. Line 272. "than in the Di Capra study"
  18. Lines 280-288.  The authors note one of the limitations as the short time of follow-up .... "may not" be able to predict the asthma development in such a short period, not "could not". Can the authors in the view of other published studies propose the likely time frame for edevleopment of asthma in PAR? Their 12 month follow-up seems a little short.

Author Response

Response to Reviewer 1 Comments

Before addressing each of the comments below, the authors would like to thank the reviewer for his/her time and valuable comments.

Point 1: Abbreviations of PAR and FeNO need to introduced early in the text after their description in full. e.g. FeNO is used several times in the introduction but only described in fill on line 104. PAR and FeNO are used in the abstract without full description, unlike HDM.

Response 1: Thank you for the observation. The authors introduced the explanation for these abbreviations in the abstract part and the explanations were marked in red in the revised manuscript.

Point 2: line 61. diagnosis not diagnose

Response 2: We corrected as you suggested (line 63)

Point 3: line 78.  Incusions/exclusions "were similar". This appears a little undefined.  Were they identical with additional exclusion as noted subsequently? I note that lines 87-88 and 89-90 and 94 suggest exclusons of  low airways symptoms, nasal obstrutions and impaired lung function by spirometry on enorllment? Is this the case?

Response 3: Thank you for this observations. We included once again the exclusion criteria in the revised version of the manuscript to be more precised (lines 82-84).

Point 4: line 98. atopy diagnosis not diagnose

Response 4: We corrected as you suggested (line 104)

Point 5: Lines 109 and 114. plasma not plasmatic

Response 5: We corrected as you suggested (line 115 and 120)

Point 6: "Quantitative data were evaluated for normality of distribution and were characterised by median and 25-75 percentiles".  The authors state they used non-parametric tests for correlations between variables. Were non-parametric comarisons used for the data in table 1 and the multivariate analyses (table 3)?  The authors could be more explicit, given they did introduce " normality of distribution".

Response 6: Thank you. Indeed we evaluated the normality of distribution and variables with abnormal distribution were characterized by median and 25-75 percentiles. The specific non-parametric tests were used for data included in table 1 (we added the information in the table's footnote). We included this explanation in the revised form of the manuscript (line 131, 133-134). Multivariate analysis was carried out using multivariate logistic regression (we added the information in the table's footnote). 

Point 7: Line 129 lose the a before cut-off values,

Response 7: We corrected as you suggested (line 135)

Point 8: Line 134 developed not develop

Response 8: We corrected as you suggested (line 141)

Point 9: It may help readers evaulate those parameters close to statistical significance if analytical precision of the methods (e.g. ICAM, VCAM, E-selection, TNF, IL6 & 8) can be provided, as analytical precision will add the the biological variation in such parameters and weaken their possible prognostic value.

Response 9:  Thank you for this comments. Because the financial support was limited to an internal grant, we did not have the possibility to calculate intra- and interassay precision coefficients. We used high sensitivity ELISA kits for all parameters that we studied and we considered the parameters provided by the manufacturer when we analyze the data. In all ELISA kits that we used the minimal detected values for our samples were higher than minimum detectable dose (MDD) obtained by manufacturer.

Point 10: Table 1 wheal mm and lines 223-227.  Are the wheal sizes for asthma/non-asthma categories in table 1 the right way around? the text and the table appear at odds.  What was the prick test antigen described in  table 1? Line 225 suggests Der p 1 and Der p 2 but the methods section for SPT suggests Derm Pterony. a Derm. Farinae?  The method section states skin prick test, but I presume a panel of skin prick tests were employed using the allergens individually as described? Maybe the authors could review and tighten up text about SPT in methods, results, table  and discussion.

Response 10:  Thank you for this comment. Yes the wheal sizes for asthma/non-asthma categories were included in table 1. We analyzed the wheal sizes of Der p allergen, which was positive in all the patients.  Sensitization for Der f was noticed only in 50 from 58 patients (86.20%). The extract that we used for Der p included both Der p 1 and Der p 2 that are considered major allergens from Derm. Pteronyssinus. The study mentioned in discussion section, published by Vidal C  et al, analyzed the level of sensitization for Der p1 and Der p2 separately though sIgE, but we analyzed the size for entire Der p extract. We change and we explain this in table 1 and in the results, discussion sections of the revised manuscript (lines 168-171, line 235)

Point 11: Table 1. Onset of AR symptoms (months). Is this duration of AR symptoms prior to enrollment? Duration is used in the text elsewhere.

Response 11:  Thank you for this comment. Yes we refer to duration of AR symptoms prior enrollment. We included this comment in material and method section, 2.2. clinical evaluation line 88.

Point 12: Lines 158-161. These are potentially important findings. Could the numbers and statistical analysis on monosentisised (i.e. HDM) versus poly-sensitisation be incorporated into table 1 to give greater clarity?

Response 12:  We considered that is not relevant for this study, because the included patients had symptoms only after exposure to HDM, and polysensitization was not actually a poly-allergy. This observation was already mentioned in the initial version of the manuscript (line 167).

Point 13: Table 2 headings Sensitivity not Sensibilty.

Response 13:  We corrected as you suggested

Point 14: Line 192. After Greisner et al, add [24].

Response 14:  We corrected as you suggested (line 203)

Point 15: Paragraphs 199-207 and 231-254. We have the concepts of phenotype and endotype introduced. I am more conversant with the former.  Is the latter, correct and necessary usage?

Response 15:  The authors considered necessary to use both terms. The terms of endotypes of rhinitis are not completely understood and defined, and the decision of treatment is still made using rhinitis’ phenotypes (which are defined according to persistence, severity of symptoms, sensitization, level of control after treatment). But analyzing other possible noninvasive biomarkers in patients with type 2 immune response in rhinitis (the endotype of rhinitis from the present study), will help us to define better different subgroups of patients with a specific phenotype of allergic rhinitis, which may require different approach (for example: patient with high risk of asthma which might require more visits, repeated FeNo measurements or even bronchial provocation test, etc). We underline both term in discussion section lines 215-216, to understand better that one term does not exclude the other one.

Point 16: Line 266. risk not risks

Response 16:  We corrected as you suggested (line 276)

Point 17: Line 272. "than in the Di Capra study"

Response 17:  We corrected as you suggested (line 283)

Point 18: Lines 280-288.  The authors note one of the limitations as the short time of follow-up .... "may not" be able to predict the asthma development in such a short period, not "could not". Can the authors in the view of other published studies propose the likely time frame for edevleopment of asthma in PAR? Their 12 month follow-up seems a little short.

Response 18:  We corrected as you suggested (line 294). Yes indeed it was a short period of surveillance, but it was the end of the project and unfortunately we cannot follow up longer all the patients after ending the projects. For this reason we decided to analyze at 1 year time point all the patients that were enrolled.

Reviewer 2 Report

The work is well constructed even if it is placed in a known context. I have made some comments and reported on the bibliography which may perhaps be added, in particular with a view to the progression of the disease from pediatric to adulthood in the introduction phase.

“It has been known for a long time that rhinitis in general, and allergic rhinitis in particular, is a risk factor for developing asthma (1) especially if the onset is severe and occurs in childhood (2).

It is also known that the upper and lower respiratory tract represent a unique functional unit (3) and that therefore the inflammation of the upper respiratory tract reflects the inflammation of the lower respiratory tract (4,5), however a high nitric oxide is more correlated. with the risk of having an asthmatic access rather than a predisposition to have asthma.(6)

Furthermore, the average age of the patients is rather high and not significant for a development of asthma at this age compared to what can be seen in children.”

Too bad that the patients are not many and that the age is not lower however I have no particular notes to make besides the above suggestions.

1 Rochat MK, Illi S, Ege MJ, Lau S, Keil T, Wahn U, von Mutius E; Multicentre Allergy Study (MAS) group

Allergic rhinitis as a predictor for wheezing onset in school-aged children.

J Allergy Clin Immunol. 2010 Dec;126(6):1170-5.e2. doi: 10.1016/j.jaci.2010.09.008. Epub 2010 Nov 4.

2 Kurukulaaratchy RJ, Zhang H, Patil V, Raza A, Karmaus W, Ewart S, Arshad SH.

Identifying the heterogeneity of young adult rhinitis through cluster analysis in the Isle of Wight birth cohort.

J Allergy Clin Immunol. 2015 Jan;135(1):143-50. doi: 10.1016/j.jaci.2014.06.017. Epub 2014 Jul 29.

3 Kanda A, Kobayashi Y, Asako M, Tomoda K, Kawauchi H, Iwai H.

Regulation of Interaction between the Upper and Lower Airways in United Airway Disease.

Med Sci (Basel). 2019 Feb 11;7(2)

4 Amorim MM, Araruna A, Caetano LB, Cruz AC, Santoro LL, Fernandes AL.

Nasal eosinophilia: an indicator of eosinophilic inflammation in asthma.

Clin Exp Allergy. 2010 Jun;40(6):867-74

5 Gelardi M, Iannuzzi L, Quaranta N, Landi M, Passalacqua G.

NASAL cytology: practical aspects and clinical relevance.

Clin Exp Allergy. 2016 Jun;46(6):785-92. doi: 10.1111/cea.12730.

6 Buhl R, Korn S, Menzies-Gow A, Aubier M, Chapman KR5, Canonica GW6, Picado C, Donica M, Kuhlbusch K, Korom S, Hanania NA.

Prospective, single-arm, longitudinal study of biomarkers in real-world patients with severe asthma.

J Allergy Clin Immunol Pract. 2020 Apr 15. pii: S2213-2198(20)30338-X. doi: 10.1016/j.jaip.2020.03.038. [Epub ahead of print]

Author Response

We want to thank you to reviewer for his/her valuable comment. Some of the suggestions were included in the introduction section (line 44-45 and 64-65)  together with the suggested references.

Indeed in asthmatic patients a high nitric oxide is more correlated with the risk of having an asthmatic access rather than a predisposition to have asthma. But the aim of this study was to analyze if an increased NO in patients with allergic rhinitis might also indicate over time the occurrence of asthma, even if the moment of evaluation the patients had no asthma symptoms.

We included only adult patients with an average age of 27.5 years old. But allergic rhinitis could also evolve to asthma in adulthood. It the present study a longer duration of allergic rhinitis in older patients might influence the occurrence of asthma symptoms in adulthood. It could be interested to evaluate the same parameters in both adults and children to see if there are differences as we suspected.